# Identifying Significantly Perturbed Subnetworks in Cancer Using Multiple Protein–Protein Interaction Networks

**DOI:** 10.3390/cancers15164090

**Published:** 2023-08-14

**Authors:** Le Yang, Runpu Chen, Thomas Melendy, Steve Goodison, Yijun Sun

**Affiliations:** 1Department of Microbiology and Immunology, The State University of New York at Buffalo, Buffalo, NY 14203, USA; lyang25@buffalo.edu (L.Y.); runpuche@buffalo.edu (R.C.); tmelendy@buffalo.edu (T.M.); 2Department of Quantitative Health Sciences, Mayo Clinic, Jacksonville, FL 32224, USA; goodison.steven@mayo.edu; 3Department of Computer Science and Engineering, The State University of New York at Buffalo, Buffalo, NY 14203, USA

**Keywords:** cancer, driver genes, driver pathways, significantly perturbed subnetworks, protein–protein interaction

## Abstract

**Simple Summary:**

Cancer research has been increasingly focusing on identifying genes and molecular pathways that drive the disease. In this context, protein–protein interaction networks, which provide insights into the interactions among proteins within a cell, have proven particularly useful. However, the effectiveness of existing approaches can be influenced by the specific network used, as different networks can have different topological structures. In addition, newer context-specific networks often come with incomplete structures, which complicates the analysis. To address these challenges, we propose a new method, called MultiFDRnet, that can identify driver genes and pathways using multiple protein–protein interaction (PPI) networks. Here, the false discovery rate (FDR) refers to the proportion of non-cancer genes within identified subnetworks. Our method, tested on both simulated and real cancer data, has been able to identify important subnetworks that are supported by multiple PPI networks and reveal novel modular structures in context-specific PPI networks. The software that we developed to implement this method is freely available for other researchers to use.

**Abstract:**

Background: The identification of cancer driver genes and key molecular pathways has been the focus of large-scale cancer genome studies. Network-based methods detect significantly perturbed subnetworks as putative cancer pathways by incorporating genomics data with the topological information of PPI networks. However, commonly used PPI networks have distinct topological structures, making the results of the same method vary widely when applied to different networks. Furthermore, emerging context-specific PPI networks often have incomplete topological structures, which pose serious challenges for existing subnetwork detection algorithms. Methods: In this paper, we propose a novel method, referred to as MultiFDRnet, to address the above issues. The basic idea is to model a set of PPI networks as a multiplex network to preserve the topological structure of individual networks, while introducing dependencies among them, and, then, to detect significantly perturbed subnetworks on the modeled multiplex network using all the structural information simultaneously. Results: To illustrate the effectiveness of the proposed approach, an extensive benchmark analysis was conducted on both simulated and real cancer data. The experimental results showed that the proposed method is able to detect significantly perturbed subnetworks jointly supported by multiple PPI networks and to identify novel modular structures in context-specific PPI networks.

## 1. Introduction

Identifying cancer driver genes and pathways is a critical endeavor in both cancer research and clinical practice [1,2,3]. While frequency-based methods have been successful in pinpointing putative driver genes by identifying genes with mutation rates higher than expected during cancer progression, these methods often struggle when it comes to identifying known driver genes that are mutated in only a small number of patients [4]. Thus, relying solely on mutation frequencies can pose challenges in distinguishing new candidate driver genes from non-driver genes [5]. Moreover, frequency-based methods often produce a list of potential driver genes lacking a cohesive theme of biological processes [6]. To address the above limitations, several network-based methods have recently been developed that integrate genomics data with topological information of protein–protein interaction (PPI) networks [5,6,7,8]. The hypothesis is that known driver genes form functional modules in a PPI network, and that genes in these modules can provide alternative targets of interest [5].

Most existing network-based methods rely on input from a single PPI network, and the topological structure of the input network plays a crucial role in the identification of candidate driver genes and module structures. With the development of molecular techniques to identify protein–protein interactions, several widely used PPI networks have been developed, including BioGRID [9], iRefIndex [10], ReactomeFI [11] and STRING [12], each with a distinct topological structure. Thus, there is a need for development of a method that is capable of identifying subnetworks that are significantly perturbed in cancer cells through the leveraging of multiple PPI networks. Another limitation of existing methods is that they cannot be applied to context-specific PPI networks (e.g., tissue- or disease-specific networks) [9,10,11,12,13,14,15] for the identification of modules in various cellular environments, as their topological information is often incomplete. For example, a context-specific PPI network generated by using cancer cell lines [14] contains only 675 genes and 1677 interactions (See Table A1). One possible way to address these issues is through pre-processing by integrating multiple PPI networks into a single network. However, the integration operation often alters the topological structures, and detected subnetworks can be significantly different from those detected on individual networks. Alternatively, post-processing can be performed, as proposed by HotNet2 [5] and Hierarchical HotNet [16], that generates consensus subnetworks by combining results obtained from individual networks. Although the consensus approaches can combine the topological information of multiple PPI networks, they suffer from two major drawbacks. First, both methods identify subnetworks in individual networks without using any information from other networks, which may result in missing structures that could be revealed by considering all PPI networks together. Second, the consensus procedures assume that every input PPI network has a complete topology covering most of the protein-coding genes [17]. However, when any input network is incomplete (e.g., context-specific PPI networks), the detected subnetworks may not be biologically meaningful.

In this study, we developed a novel algorithm, referred to as MultiFDRnet, for the detection of significantly perturbed subnetworks using multiple PPI networks. It is a natural extension of our previously developed FDRnet method but incorporates two key improvements that address the aforementioned limitations. First, it is able to detect significantly perturbed subnetworks supported by multiple, widely used PPI networks, thereby eliminating the need to perform pre- or post-processing procedures. Second, it can reveal perturbed modular structures within context-specific networks by effectively using the topological information from general PPI networks as a complementary resource. To achieve the above improvements, we employed the multiplex-network framework. With its power and flexibility, the framework can model multiple PPI networks, with or without emphasizing one or more networks. To detect subnetworks within a multiplex network, we developed a novel random-walk strategy to quantify subnetworks. To illustrate the effectiveness of our proposed method, we conducted a large-scale numerical study on both simulated and cancer mutational data. The developed software and data are freely available at https://github.com/yangle293/MultiFDRnet (accessed on 13 August 2023).

## 2. Materials and Methods

MultiFDRnet takes as input multiple PPI networks and a set of *p*-values derived from gene-level analyses (e.g., MutSig [4]) that measure the likelihood that observed mutation rates are due to a random chance, and outputs a collection of significantly perturbed subnetworks (Figure 1). Briefly, we first construct a multiplex network using input networks (Figure 1a). Subsequently, we perform an empirical Bayes analysis that uses the *p*-values to determine the probability, or local FDR, of each gene being unrelated to cancer (Figure 1b). We designate genes that have local FDRs below a specified FDR bound as seeds. For each seed, a random walk-based approach is employed to define the optimal subnetwork around this seed (Figure 1c). Finally, the problem of searching for the optimal subnetworks, given an FDR bound, is formulated as a mixed-integer linear programming problem. This results in the identification of significantly perturbed subnetworks in multiplex networks (Figure 1d). In the sections that follow, we present a detailed description of our method.

### 2.1. Modeling Multiple PPI Networks as a Multiplex Network

We model a set of PPI networks as a multiplex network [18,19] to preserve the topological structures of individual networks and dependencies among them. Mathematically, a multiplex network is defined as a quadruple $M=(V_{M},E_{M},V,L)$, where *V* is a set of physical nodes, *L* is a set of layers, $V_{M}$ is a set of state nodes, defined as the Cartesian product of *V* and *L* (i.e., $V_{M}=V\times L$), and $E_{M}$ is a set of edges between state nodes, $E_{M}\subseteq V_{M}\times V_{M}$. Essentially, a multiplex network is a network defined by $V_{M}$ and $E_{M}$ with a layer structure. In the rest of this paper, we use i∈V to denote the physical node and $i\alpha \in V_{M}$ to denote the state node that represents node *i* on layer α.

To construct a multiplex network, we set genes in a PPI network as physical nodes and treat each network as a layer. Specifically, given a set of networks $G=\{G_{\alpha }|\alpha \in I\}$, where $G_{\alpha }=\left(V_{\alpha },E_{\alpha }\right)$, we set L=I and $V={⋃}_{\alpha \in L}V_{\alpha }$. The set of state nodes is built as $V_{M}=V\times L$. Note that the dependencies across layers are automatically introduced into the multiplex network, since state nodes from different layers represent the same set of physical nodes. For each layer α, we copy the edges from $G_{\alpha }$ into the layer α of the multiplex network M. That is, $E_{M}={⋃}_{\alpha \in L}\left\{\left(i\alpha ,j\alpha \right)|\forall \left(i,j\right)\in E_{\alpha }\right\}$. If we need to emphasize the structure of a network $G_{\beta }$, we can set the weights of all the edges in the layer β as a user defined parameter w>1, meaning that we focus on the structure of a specific network; otherwise, all edge weights are set to 1. In this way, we preserve all the topology information in $E_{M}$.

### 2.2. Defining False Discovery Rate for Subnetworks in Multiplex Networks

Next, we define the false discovery rate (FDR) for subnetwork identification in a multiplex network. FDRnet proposed a definition of FDR for subnetwork identification based on the fact that genes, rather than subnetworks, are the smallest units in statistical analysis [6]. FDRnet initially conducts an empirical Bayes analysis to estimate local FDRs for individual genes based on their *p*-values. The FDR of a subnetwork can then be derived by averaging the local FDRs of the genes present in the subnetwork. With MultiFDRnet, this concept is extended to subnetworks in a multiplex network. To start, each gene, interpreted as a physical node, is assigned a local FDR score using the same analysis as in FDRnet. Subsequently, the FDR of a subnetwork is computed by averaging the local FDR scores of the physical nodes associated with the subnetwork. Here, we say a physical node is associated with a subnetwork if it can be represented by one or more state nodes in this subnetwork. Mathematically, for a subnetwork S⊆M, we have $\mathrm{FDR}\left(S\right)=1/|P_{S}|\sum_{j\in P_{S}\subset V}w_{j}$, where $P_{S}=\left\{i\in V|\exists i\alpha \in S,\alpha \in L\right\}$ and $w_{j}$ is the local FDR score of gene j∈V. We control the FDR of detected subnetworks by requiring the FDR to be less than a given bound *B*. To identify subnetworks, we select, as seeds, the genes with a local FDR less than a given bound *B* and identify a subnetwork for each seed.

### 2.3. Random Walk-Based Approach to Subnetwork Identification

Our goal is to search an optimal subnetwork around each seed with its FDR being controlled. To this end, we propose a random walk-based approach to quantify subnetworks in the multiplex network. A random walk, when defined in the context of a multiplex network, stands as a dynamic process capable of navigating within and between layers. This characteristic of the random walk makes it an effective tool for harnessing the topological structures and dependencies embedded in input PPI networks. Mathematically, a random walk in a multiplex network is specified by a transition matrix T, where $T_{i\alpha ,j\beta }$ is the transition probability for a random walker from state node iα to jβ. In the following, we quantify a subnetwork in the multiplex network from three aspects.

First, based on the assumption that driver genes are clustered in the network, we aim to identify cluster-like subnetworks in the multiplex network. In such a cluster-like subnetwork, the dynamic process characterized by a random walk is expected to exhibit a bottleneck feature, which implies that random walkers are more likely to remain confined within the subnetwork. To quantify this bottleneck effect, we employ the generalized conductance score. This metric is defined as the outflow of random walkers from a subnetwork $S=(V_{S},E_{S})$ relative to the total number of random walkers within *S* at stationary [20]:(1)$$\Phi \left(S\right)=\sum_{i\alpha \in V_{S}}\sum_{j\beta \notin V_{S}}T_{i\alpha ,j\beta }p_{i\alpha }/\sum_{i\alpha \in V_{S}}p_{i\alpha },$$
where $p_{i\alpha }$ is the stationary distribution of the random walker at state node iα, satisfying $p_{i\alpha }=\sum_{j\beta \in V_{M}}T_{j\beta ,i\alpha }p_{j\beta }$. As such, our goal is to identify the subnetwork that minimizes its conductance score. However, a general subnetwork can consist of state nodes from only some layers; therefore, its conductance score cannot reflect the structures of all PPI networks. To address this issue, we impose a constraint on the identified subnetwork, denoted as a cover constraint. This constraint ensures that the subnetwork, denoted as *S*, spans all layers of the multiplex network, and the state nodes within each layer represent the same set of physical nodes. In other words, there exists a set of physical nodes, denoted as $P_{S}$, satisfying the condition $V_{S}=P_{S}\times L$.

Second, although the conductance score proves to be effective in identifying cluster-like structures, it has an inherent limitation in that it does not necessarily ensure the internal denseness of a subnetwork, even when the subnetwork displays a low conductance score. A common strategy to mitigate this limitation is to search subnetworks locally, that is, first extract a densely connected local graph and then search subnetworks inside the identified local graph [6,21,22]. In this study, we employed the approximate Personalized PageRank (PPR) algorithm [21] to detect a local graph surrounding a specific seed node. Subsequently, we constrained the search for the optimal subnetwork to be strictly within this identified local graph. Mathematically, we let the dynamic process, represented by a given random walk, start from all state nodes associated with a seed node *s* by initializing a residual vector r with length |V|×|L| as $r_{s\alpha }=1/\left|L\right|,\forall \alpha \in L$ and 0 otherwise. Then, we used the approximate PPR algorithm to push probability mass from the residual vector r to the PageRank score vector p by using transition matrix T until the residual was smaller than a given threshold ϵ. We repeated this procedure, each time decreasing the value of ϵ, as long as the number of non-zero entries in vector p exceeded a predefined exploration size *K*. If this condition was not met, the process was stopped. After the process was completed, we retained the state nodes that had the highest *K* PageRank scores. To ensure all the layers were equally covered, we augmented the local graph by adding state nodes iα,∀α∈L if other state nodes, representing the same physical node *i* (e.g., iβ), were already included. There are two parameters, the teleportation parameter in the approximate PPR algorithm and the local exploration size *K*. As in [21], we fixed the teleportation parameter as 0.998. We later show that the performance of our method is not sensitive with respect to *K*. Across all the experiments, we set *K* = 400 as the default parameter.

Finally, we aimed to ensure that each identified subnetwork was connected in the layered structure of the multiplex network. To accomplish this, we used a random-walk method to define this kind of connectivity. We started by constructing a graph with an adjacent matrix $T^{adj}$. This matrix is defined as $T_{i\alpha ,j\beta }^{adj}=1$ if $T_{i\alpha ,j\beta }>0$ and $T_{i\alpha ,j\beta }^{adj}=0$ otherwise. This approach was motivated by the idea that, in terms of a random walker, if it is possible to transition from iα to jβ, these two nodes should be connected. In this manner, we could simplify the issue of multiplex network connectivity to the problem of standard graph connectivity, which is well-studied within optimization theory [23].

There are several ways to define a random walk in a multiplex network, each providing a unique approach to navigate its layered structure [24,25,26]. For our study, due to its simplicity, we chose to use the classic random walk [24,25] to calculate conductance score, extract local graphs and define connectivity. For the sole purpose of calculating the transition probability matrix, we added interlayer edges connecting state nodes representing the same physical nodes across different layers, i.e., $E_{inter}=\left\{\left(i\alpha ,i\beta \right)\right|\forall i\in V,\forall \alpha ,\beta \in L$ }. The transition matrix of the classic random walk is then calculated as $T_{i\alpha ,j\beta }=A_{i\alpha ,j\beta }/\sum_{j\beta \in V_{M}}A_{i\alpha ,j\beta }$, where A is the adjacent matrix (weighted if we define layer weight other than 1) after adding $E_{inter}$.

### 2.4. Identifying Subnetworks Using Mixed-Integer Linear Programming

Putting all the above together, we formulated the subnetwork identification problem as follows:

**Problem** **1.**
*Given a multiplex network $M=(V_{M},E_{M},V,L)$ with layer weight w:L→R, a seed s, a local graph $Local_{s}$ around s, a transition matrix T and a budget B, find a subnetwork $S\subseteq Local_{s}\subseteq M$, that satisfies budget and cover constraint, is connected in the graph defined by the adjacent matrix $T^{adj}$ and minimizes Φ(S).*


By defining a binary variable $x_{i\alpha }$ for each state node $i\alpha \in V_{M}$, which signifies whether iα is included in a solution, we can convert the above problem into an integer programming problem:(2)$$\begin{array}{cc} \mathrm{minimize}\; & \sum_{i\alpha \in V_{M}}\sum_{j\beta \in V_{M}}x_{i\alpha }\left(1-x_{j\beta }\right)T_{i\alpha ,j\beta }p_{i\alpha }/\sum_{i\alpha \in V_{M}}p_{i\alpha }x_{i\alpha }\\ \mathrm{subject}\;\mathrm{to}\;\; & \frac{1}{\sum_{i\in V}x_{i\alpha }}\sum_{i\in V}x_{i\alpha }w_{i}\leq B,\;\;\;\;\;\left(\mathrm{FDR constraint}\right)\\ \; & x_{j\beta }\in \left\{0,1\right\},\forall j\beta \in V_{M},\;\;\;\;\;\;\left(\mathrm{binary constraint}\right)\\ \; & x_{s\alpha }=1,\;\;\;\;\;\;\;\;\;\;\;\;\left(\mathrm{seed constraint}\right)\\ \; & x_{i\beta }=x_{i\gamma },\forall i\in V,\forall \beta ,\gamma \in L,\beta \neq \gamma ,\;\;\left(\mathrm{cover constraint}\right)\\ \; & V_{S}=\{j\beta :x_{j\beta }=1,j\beta \in V_{M}\}\;\;\mathrm{forms}\;a\;\mathrm{connected}\;\mathrm{subgraph},\left(\mathrm{connectivity constraint}\right)\\ \; & x_{j\beta }=0,\forall j\beta \notin Local_{s}.\;\;\;\;\;\;\;\left(\mathrm{local graph constraint}\right) \end{array}$$

We employed the strategies used by FDRnet [6] to recast the problem as a mixed-integer linear programming problem, which can be solved efficiently using established optimization tools, such as CPLEX [27]. We present the details of the linearization procedures in Appendix A. We solved a linear programming problem for each seed to detect a densely connected subnetwork. To eliminate potential redundant subnetworks, we arranged the acquired seeds in descending order, based on the count of other seeds in their immediate vicinity in the aggregated network, and skipped a seed if it had already been incorporated in a subnetwork identified earlier.

## 3. Results

To demonstrate the efficacy of our proposed method, extensive numerical studies were carried out using both simulated and actual cancer data. We compared it with the following six other methods: FDRnet [6], HotNet2 [5], hierarchical HotNet [16], Netmix2 [8], BioNet [28,29] and Domino [7].

### 3.1. Simulation Study

Network-based methods take as input one or several PPI networks and a set of *p*-values and output a list of significantly disrupted subnetworks. In this study, we used four PPI networks, including BioGRID [9], iRefIndex [10], ReactomeFI [11] and STRING [12], which are widely used in network analysis [30,31,32]. The networks were considered as unweighted and undirected, since the majority of the existing methods were developed for this particular scenario. To remove low-quality interactions in the STRING network, we followed the common practice to retain interactions with a confidence score larger than 900 [30]. Among all the competing methods, MultiFDRnet, HotNet2 and hierarchical HotNet can take multiple PPI networks as input, while the other algorithms can only take one network. In order to make a fair comparison, we constructed a new PPI as input, denoted as AggrePPI, by aggregating all the interactions of the four PPI networks. We also tried a more advanced network integration algorithm BIONIC [33], but it was unable to generate an integrated network for the four PPI networks above in a workable timeframe (≥72 h). The information of the PPI networks used in this study is summarized in Table A1 and Figure A1.

To generate synthetic *p*-values, we followed the procedures described in [6]. Specifically, we selected 16 protein complexes from the CORUM database [34] as our target subnetworks. These protein complexes, which contain between 10 to 50 proteins, are known to play roles in the progression of breast cancer. We employed the signal-to-noise decomposition model [29,35] to generate synthetic *p*-values. This model posits that the distribution of *p*-values consists of two parts: one stemming from the null hypothesis and the other from the alternative hypothesis. It is conceivable that *p*-values derived from the null hypothesis uniformly span the range (0,1). Under the alternative hypothesis, the distribution of *p*-values is characterized by a high density at lower values, which progressively diminishes as the *p*-values increase. This pattern aligns well with a specific form of the beta distribution: Beta(a,1). Therefore, the *p*-values for target genes (i.e., genes within target networks) were randomly selected from a beta-distribution Beta(a,1), and the *p*-values for non-target genes were sampled from a uniform distribution U(0,1). To evaluate the performance of a method when applied to data with varying signal strengths, we adjusted the values of *a* to a range of 0.01 to 0.11 in increments of 0.01, with smaller values signifying greater strength. For each *a*, we performed the experiment 10 times to minimize random fluctuations. For MultiFDRnet, FDRnet, Netmix2 and BioNet, we used the synthetic *p*-values directly for input. For HotNet2 and hierarchical HotNet, following the instructions given in [16,36], we used $-\log_{10}\left(q\right)$ as input and the consensus results of the four PPI networks as output. Here, *q* is the adjusted *p*-value computed by the Benjamini–Hochberg procedure [37]. For Domino, we used the aggregated PPI as input and set, as seeds, the genes with a local FDR score less than 0.1.

For MultiFDRnet and FDRnet, we set the FDR upper bound *B* to 0.1, and the local exploration size (i.e., the load graph size) *K* to 400. We selected, as seeds, genes with a local FDR score below 0.1. For each seed, we solved a mixed-integer linear programming problem (Equation 2) to identify a subnetwork. In Section 3.1.3, we performed a parameter sensitivity analysis and demonstrated that our method is largely insensitive to the specific choice of *K*. We set the FDR parameter τ to 0.1 for BioNet. All other parameters were set to default values. All the experiments were performed at the University at Buffalo high-performance research computing center using 16 × 2.20 GHz Intel E5-2660 Xeon Processor Cores (Manufacturer: Dell Inc., Round Rock, TX, USA) and 128 GB memory.

#### 3.1.1. Evaluation Metrics

Three metrics were used to evaluate the effectiveness of an algorithm in identifying target genes, detecting target subnetworks, and controlling FDRs. The *F* score [6,38], calculated by comparing the list of genes in target subnetworks with those in detected subnetworks, was used as a measure of the ability of a method to identify target genes. A symmetric version of the $F_{\mathrm{sub}}$ score, proposed in [6], was employed to assess the capability of a method to detect target subnetworks. Specifically, the symmetric $F_{\mathrm{sub}}$ score between a set of *M* subnetworks $A=\{A_{1},\cdots ,A_{M}\}$ and another set of *N* subnetworks $B=\{B_{1},\cdots ,B_{N}\}$ is defined as:(3)$$F_{\mathrm{sub}}\left(A,B\right)=\frac{1}{2}\left(\frac{1}{\sum_{i}\left|A_{i}\right|}\sum_{i=1}^{M}|A_{i}|\max_{j\in \{1,\cdots ,N\}}F\left(A_{i},B_{j}\right)\;+\frac{1}{\sum_{i}\left|B_{i}\right|}\sum_{i=1}^{N}\left|B_{i}\right|\max_{j\in \{1,\cdots ,M\}}F\left(B_{i},A_{j}\right)\right)\;,$$
where $F(A_{i},B_{j})$ is the *F* score between $A_{i}$ and $B_{j}$. Finally, to assess how well an algorithm controlled the false discovery rates of detected subnetworks, the FDR definition proposed in [6] was employed. Specifically, the FDR of an identified subnetwork was determined by the proportion of non-target genes present in that subnetwork. We also reported the estimated FDRs, as defined in Section 2.2. The running time was recorded to compare the computational complexities.

#### 3.1.2. Experimental Results

First, we evaluated the capabilities of the seven methods to identify target genes (by *F*-scores) and modular structures (by symmetric $F_{\mathrm{sub}}$ scores). Figure 2a,b presents the results of the seven methods when applied to synthetic data generated by using a range of beta-distribution parameters *a*, varying from 0.01 to 0.11. In terms of *F*-scores, MultiFDRnet, FDRnet, and BioNet had the best performance, significantly exceeding all other methods. When considering symmetric $F_{\mathrm{sub}}$ scores, MultiFDRnet notably surpassed all other methods, while BioNet performed poorly in this regard. BioNet performed well in terms of *F*-score, since the distributions of synthetic *p*-value perfectly matched the assumption used in BioNet. However, it performed poorly in terms of symmetric $F_{\mathrm{sub}}$ score. This is because it connected all the genes into one large subnetwork. With the decrease of signal strengths (i.e., the increase of the value of *a*). We observed the following: (1) the performance of all the methods declined, and (2) in nearly all instances, the relative performance rankings of the different methods remained the same.

To conduct a further comparison between MultiFDRnet and FDRnet, we applied FDRnet to individual PPIs and report the results in Figure 2c,d. We can see that, in terms of *F*-scores, MultiFDRnet performed at least as well as FDRnet applied to individual PPI networks or the aggregated network. However, in terms of symmetric $F_{\mathrm{sub}}$ scores, MultiFDRnet performed significantly better than FDRnet applied to individual PPI networks. This result suggests that the proposed multiplex strategy can improve the ability to detect modular structures. In contrast, FDRnet applied to the aggregated network performed worse than FDRnet applied to some of the individual PPI networks. One possible explanation is that the aggregation operation alters the topological structures, which may compromise the ability of FDRnet to detect target genes and modular structures. We also investigated the effectiveness of the consensus procedure used in HotNet2 and hierarchical HotNet. To this end, we compared the subnetworks obtained by using the consensus procedure with those obtained by applying HotNet2 and hierarchical HotNet to individual PPI networks. The comparison results are reported in Figure A2 and Figure A3. We found that the consensus procedure did not improve, but rather compromised the abilities of HotNet2 and hierarchical HotNet to detect target genes and modular structures.

We then assessed the capabilities of the seven methods to control the FDRs of identified subnetworks. Figure 3 and Figure A4 report the estimated FDRs and exact FDRs of the subnetworks detected at varying *a* levels. We can see that, except for Domino, all the algorithms could control FDRs to some extent. Specifically, for MultiFDRnet and FDRnet, the estimated FDRs of detected subnetworks were controlled at the level of 0.1, as expected. For the simulation data, exact FDRs could be calculated by utilizing ground-truth information. We found that, for MultiFDRnet and FDRnet, the exact FDRs of small subnetworks detected could be larger than 0.1. This is due to small subnetworks being more vulnerable to noise. BioNet also successfully controlled the FDRs of the detected subnetworks. Nevertheless, as previously mentioned, the *p*-value distribution aligned perfectly with the assumption employed in BioNet. This might not hold true for cancer data, as we see shortly. For HotNet2 and hierarchical HotNet, since they both performed a consensus procedure, we compared the FDRs of the consensus subnetworks with those of the subnetworks detected in individual PPIs. Figure A5 and Figure A6 report the results for HotNet2 and hierarchical HotNet, respectively. We determined that the FDRs of the subnetworks detected in individual PPIs were not controlled at the desired level, while the FDRs of subnetworks obtained through the consensus procedure were. This suggests that the consensus procedure could help to control FDRs. However, as demonstrated in Figure A2 and Figure A3, this was achieved at the expense of the ability to identify target genes and modular structures. Finally, we observed that Netmix2 could control the FDRs at a level around 0.25. This was possible due to the fact that Netmix2 employs a strategy similar to that used in FDRnet and the proposed method to control FDRs. However, Netmix2 does not provide users with a parameter to adjust the FDR level.

Lastly, we assessed the computational complexities (i.e., computational time required) of the seven methods (Table 1). Domino was the fastest method, FDRnet second, followed by MultiFDRnet, hierarchical HotNet, BioNet, Netmix2 and HotNet2. Domino works by first clustering the entire network using fast heuristics and then identifying clusters that are enriched with highly mutated genes. However, there is no guarantee that the ground truth clusters are included in the initial clustering results, which can lead to suboptimal results, as shown in Figure 2a,b. Among the two best performing methods, MultiFDRnet and FDRnet, it was observed that MultiFDRnet generalized FDRnet to a more complex multiplex structure, but did not significantly increase the computational complexity.

#### 3.1.3. Parameter Sensitivity Analysis

The proposed method has two parameters, namely local exploration size *K* and user-defined FDR upper bound *B*. The local exploration size *K* is used to extract a subgraph around a given seed to provide a rough solution for conductance minimization. Thus, the value of *K* within a wide range should have little impact on the performance of the algorithm. We found that our method performed similarly for different values of *K*, ranging from 100 to 500, in terms of *F*-score, and slightly worse for K=500, in terms of symmetric $F_{\mathrm{sub}}$ score (Figure A7). Therefore, we set K=400 as a default parameter in all other experiments in the study. The user-defined FDR upper bound *B* was used to define a tolerance level of false positives. Theoretically, when the signal is strong, a tighter bound should be beneficial, since the majority of false positives are removed. In contrast, when the signal is weak, a looser bound allows the selection of true genes that cannot be distinguished with noise, leading to better total performance. To test whether this was the case for our approach, we set local exploration size *K* to 400 and performed an experiment using B∈{0.1,0.15,0.2,0.25}. We observed that the result confirmed our reasoning (Figure A8). However, in real applications, we may not have prior knowledge of signal strength, and a FDR level of 0.1 is a safe choice.

We also performed an experiment to assess how much the proposed method relies on the reliability of individual networks. To this end, we first generated a random network for each of the four PPI networks, by randomly swapping the edges of each node, while maintaining the original degree distributions. The same procedure was also used in HotNet2 [5]. Then, we repeated the simulation study by replacing one of the four PPI networks with its random counterpart. The results are presented in Figure A9. As we expected, the performance of our method declined. However, it was very robust against variations from individual networks. Notably, it performed significantly better than other approaches that were applied to AggrePPI constructed by using the original four PPI networks (see Figure 2a,b).

### 3.2. Bladder Cancer Study

The TCGA copy number and mutational data for bladder cancer was subjected to MultiFDRnet to detect significantly mutated subnetworks.

#### 3.2.1. Mutational Data and PPI Networks

The somatic mutation and copy number data were downloaded from the TCGA firehose website. The TCGA mutation data includes 94,534 non-silent mutations across 18,295 genes from 395 patients with bladder cancer. The data was evaluated by using MutSig2CV [4], where each gene was given a *p*-value indicating its statistical significance of mutation frequency in contrast to a background mutation model. The copy number data, analyzed by GISTIC2 [39], details the copy numbers of 24,776 genes in 411 individuals diagnosed with urothelial bladder carcinoma. As with the simulation study, we used four PPI networks in the analysis. We used the pipeline described in [6] to obtain three *p*-values for each gene, measuring the statistical significance of mutation, copy number amplification and copy number deletion, respectively. For the methods that use *p*-values as input (BioNet, hierarchical HotNet, HotNet2, and Netmix2), we combined *p*-values for each gene using Fisher’s method [40]. For other methods, we retained minimum local FDR scores for each gene, since local FDR scores are comparable across different data types [6]. For all methods, we used the same setting as in the simulation study.

#### 3.2.2. Experimental Results

By applying MultiFDRnet to the bladder cancer data, a total of 77 genes and 24 subnetworks were detected. For comparison, we applied the other six methods to the data. The experimental settings were similar to those used in the simulation study. Table 2 summarizes the detected subnetworks and the number of detected genes that were also included in the COSMIC database [41]. Figure 4 and Figure A10 present the subnetworks detected by MultiFDRnet and the estimated FDRs, respectively. We made the following observations. First, BioNet failed to identify any subnetworks, since the distribution of *p*-values could not be fit by its signal-to-noise model. Second, all the methods, except for Domino, could control FDRs to some extent, which was consistent with the result of the simulation study. However, HotNet2, hierarchical HotNet and Netmix2 grouped all identified genes into one subnetwork and did not reveal any modular structure. Finally, MultiFDRnet and FDRnet performed similarly, in terms of the numbers of the detected genes and subnetworks. However, a close examination of the results showed that FDRnet missed three well-established cancer genes (*KRAS*, *PIK3CA* and *PTEN*, Figure A11). Interestingly, we found that we were able to detect all three genes if we applied FDRnet to any of the individual networks. This result suggests that aggregation could lead to suboptimal results, as it alters the topological structures of individual networks.

To annotate the subnetworks identified by MultiFDRnet, we compared the genes detected in each subnetwork with the hallmark and curated gene sets provided by the molecular signatures database [42,43] and annotated the subnetwork using the label returned by the database. If a gene appeared in two detected subnetworks, we added a crosstalk edge between the two subnetworks, which implies that the gene may be involved in two biological processes. The first notable subnetwork was a densely connected subnetwork formed by *KRAS*, *HRAS*, *ERBB2*, *FRS2*, *FGFR3*, *GOLGA7* and *ERLIN2*. All genes, except for *ERLIN2*, came from the MAPK family signaling cascade, which is an important therapeutic target in cancer [44]. Similarly, we found subnetworks dominated by the PIK3CA signaling pathway and the PTEN regulation pathway. Moreover, a subnetwork was dominated by genes related to the RNA polymerase II elongation process. Recently, researchers have observed that many cancers have widespread defects in mRNA transcription elongation [45]. Finally, the proposed method identified a subnetwork that included a well-known cancer gene *FOXA1* and four genes from the Vitamin D receptor pathway, which suggests a potentially functional connection.

### 3.3. Head and Neck Cancer Study

MultiFDRnet was utilized to detect significantly mutated subnetworks in head and neck cancers using the TCGA copy number and mutational data. Unlike the bladder cancer study, we, herein, attempted to detect perturbed modular structures formed by the genes and interactions from a PPI network constructed specifically for head and neck squamous cell carcinoma [14].

#### 3.3.1. Mutational Data and PPI Networks

The somatic mutation and copy number data were downloaded from the TCGA firehose website.

The TCGA mutation dataset comprises 75,930 non-silent mutations in 18,291 genes from 510 individuals diagnosed with head and neck cancer. By using MutSig2CV [4], each gene was assigned with a *p*-value, indicating its statistical significance of mutation frequency against a background mutation model. Copy number data, analyzed by GISTIC2 [39], details the copy numbers for 24,776 genes in 525 patients with head and neck cancer. The HNSC PPI network was constructed by performing an affinity purification-mass spectrometry analysis in head and neck squamous cell carcinoma cell lines with 33 protein baits [14]. The network contains only 675 genes and 1677 interactions and cannot describe the complete topology of protein interactions in head and neck cancer. Therefore, we incorporated the four general-purpose PPI networks used in the bladder cancer study to aid the discovery of modular structures. As with the bladder cancer study, for the methods that take only one network as input, we aggregated all the PPI networks into one, denoted as aggrePPI-HNSC. The information on the PPI networks used are presented in Table A1 and Figure A1.

#### 3.3.2. Experimental Results

We applied MultiFDRnet and the other six methods to the mutational data. For MultiFDRnet, we used all five PPI networks and focused on the HNSC PPI network, by setting the weight of the corresponding layer to 5 and the others to 1. In addition, for both MultiFDRnet and FDRnet, we restricted the seeds to genes that appeared in the HNSC PPI network and solved a mixed-integer linear programming problem (Equation 2) for each seed. Other experimental settings were similar to those used in the bladder cancer study. Table 2 reports the subnetworks detected by the seven methods and Figure A12 presents the estimated FDRs of the detected subnetworks. Note that the COSMIC genes were not reported, since, in this study, we focused on identifying novel structures in the HNSC PPI network. Consistent with the results obtained in the bladder cancer study, BioNet failed to produce any results, Domino failed to control the FDRs, and both HotNet and Netmix2 detected only one subnetwork. Although hierarchical HotNet identified two subnetworks, one with 27 genes and the other with 6 genes, only five HNSC interactions were identified and none of them were specific to the HNSC network, suggesting that the modular structure of the HNSC PPI was not found (Figure A13).

To compare MultiFDRnet and FDRnet in depth, we also applied FDRnet to the HNSC PPI network only. The results of MultiFDRnet, FDRnet on AggrePPI-HNSC and FDRnet on HNSC are reported in Figure 5, Figure A14 and Figure A15, respectively. The subnetworks detected by MultiFDRnet were annotated in the same way as in the bladder cancer study. When applied to the HNSC network, FDRnet detected seven subnetworks, in which one had 9 genes, including *TP53*, while the others had 4 or less genes. We can see that the results were fragmented, possibly due to the incomplete topology of the HNSC network. Interestingly, we observed that MultiFDRnet combined or completed these fragmented subnetworks into functional modules. To see this, first note that MultiFDRnet identified a densely connected subnetwork, including most of the genes (7 out of 9) of the aforementioned *TP53*-related subnetwork. Although the backbone of this subnetwork was formed by edges from the HNSC network, its dense structure was completed by edges from other general PPI networks. Second, the subnetwork consisting of *MAPK1* and *HLA-A* was completed by adding the *HLA-B* gene, which is a close family member and is also high mutated. Finally, the clique consisting of *PIK3CA*, *PIK3R1* and *CCND1* was combined with known cancer genes, such as *TP53*, *MDM2* and *ERBB2*, by edges from general networks. Another example of combination was the subnetwork combined by a module consisting of *LSG1*, *CASP3* and *SF3B2* and a module consisting of *RB1*, *B2F1*, *RCL1* and *MRPL47*. These examples demonstrated that MultiFDRnet was able to use general PPI networks as complementary interaction information to identify perturbed modular structures in the HNSC PPI network. By contrast, when applied to the aggrePPI-HNSC network, FDRnet detected only four subnetworks that included interactions from the HNSC PPI network. Among these, the two larger ones did not include modular structures formed by edges from the HNSC network, and the two smaller ones each included only one HNSC-specific interaction. This was possibly due to the fact that the topological information of the disease-specific network was overwhelmed by the general-purpose PPI networks, since the former had far fewer nodes and edges. To investigate if increasing edge weights could mitigate this issue, we set the edge weights of all edges from the HNSC network in the aggrePPI-HNSC network to 5 and applied FDRnet to it; however, no improvement was observed (Figure A16).

To investigate how the layer weight affected the resulting subnetworks, we performed an experiment where we varied the layer weight by 20% (i.e., set *w* to 4 or 6) and calculated the *F*-score and symmetric $F_{sub}$ score by comparing the subnetworks detected by setting *w* to 5 with those detected by setting *w* to 4 or 6. We found that there were only minor differences (*F*-score = 0.96 and symmetric $F_{sub}$ = 0.94 between w=4 and w=5, *F*-score = 0.96 and symmetric $F_{sub}$ = 0.90 between w=6 and w=5).

## 4. Conclusions

In this paper, we introduced MultiFDRnet, a novel methodology devised for the detection of significantly perturbed subnetworks utilizing multiple PPI networks. MultiFDRnet addresses several limitations of existing network-based methods. By employing a multiplex-network framework to model a set of input PPI networks, we ensured preservation of the topological structures of all PPI networks, while introducing interdependencies amongst them. The framework also offers the flexibility to emphasize the structure of a specific PPI network, such as, for example, context-specific networks, by adjusting the corresponding weight. Furthermore, we developed a novel random walk-based approach that enables effective utilization of the topological structures and dependencies stored within the multiplex network during subnetwork detection. Our experiments demonstrated that MultiFDRnet can effectively detect significant subnetworks supported jointly by multiple PPI networks and can uncover novel modular structures within context-specific PPI networks.

While MultiFDRnet has proven effective in addressing the issues inherent in previous methodologies, there are several directions that we can pursue to further improve its performance. As part of our future work, we plan to extend the use of the multiplex network framework to integrate other types of interactions, including gene regulatory networks and metabolic relationships. This could provide a broader and more comprehensive view of cellular processes, leading to the discovery of previously unrecognized driver genes and key molecular pathways. In addition, we plan to incorporate more advanced computational models, such as graph neural networks, to further utilize the structural information within the multiplex network. We anticipate that these advanced models could enhance our ability to detect and interpret complex patterns within the data; thereby, increasing the analytical depth and accuracy of cancer gene detection.

## Figures and Tables

**Figure 1 cancers-15-04090-f001:**
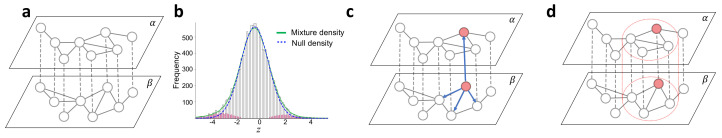
Overview of the proposed MultiFDRnet method. (**a**) Constructing a multiplex network using multiple PPI networks. (**b**) Estimating local FDR scores through an empirical Bayes analysis. Red vertical bars are estimated counts of non-null genes. (**c**) Performing random walk to quantify subnetworks. Red nodes represent two state nodes corresponding to a seed gene and blue arrows represent the possible random walks originating from one state node. (**d**) Detecting significantly perturbed subnetworks for given seeds by solving mixed-integer linear programming problems. An example solution is the subnetwork within the red circles.

**Figure 2 cancers-15-04090-f002:**
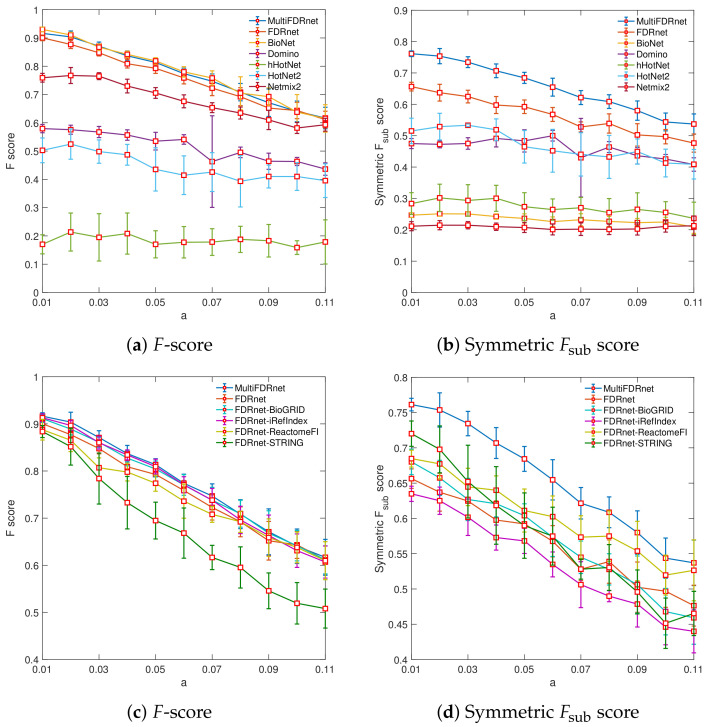
*F* scores and symmetric $F_{\mathrm{sub}}$ scores of seven methods applied to simulation data as a function of beta-distribution parameter *a*. (**a**,**b**) Comparison of MultiFDRnet with six alternative methods. (**c**,**d**) Comparison of MultiFDRnet with FDRnet when applied to aggregated and individual PPI networks.

**Figure 3 cancers-15-04090-f003:**
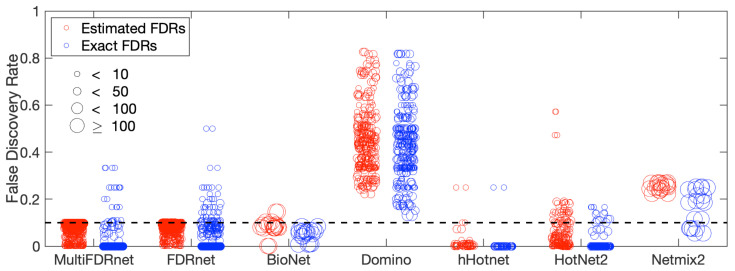
Exact FDRs and estimated FDRs of subnetworks identified by seven approaches applied to simulation data, derived from a beta-distribution with parameter *a* set at a value of 0.11. Each circle symbolizes an identified subnetwork, with its size being linearly proportional to the number of genes in the subnetwork. The FDR upper threshold was set to 0.1, marked by a dashed line.

**Figure 4 cancers-15-04090-f004:**
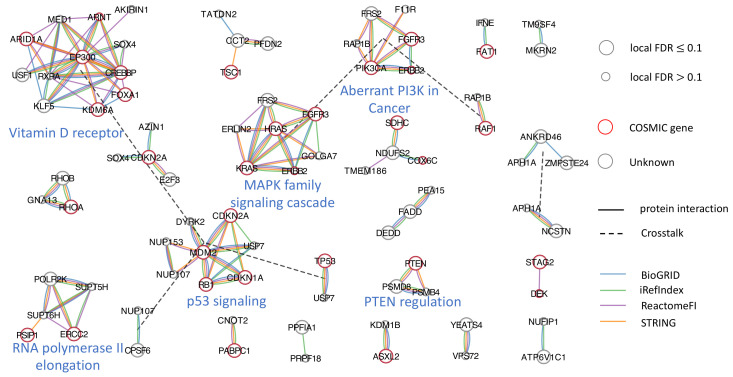
Twenty-four subnetworks detected by MultiFDRnet performed on bladder cancer data.

**Figure 5 cancers-15-04090-f005:**
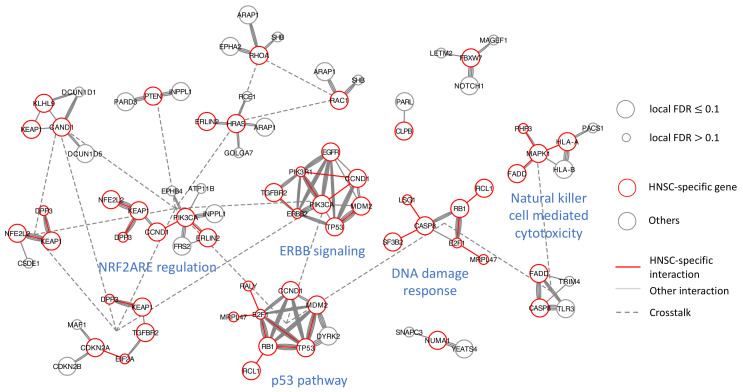
Sixteen subnetworks detected by MultiFDRnet performed on head and neck cancer data.

**Table 1 cancers-15-04090-t001:** Running time (in seconds) of seven methods applied to simulation (*a* = 0.06), bladder cancer and head and neck cancer data. For simulation data, the experiment was performed 10 times, and we reported the average execution time along with the standard deviation. HHotNet: Hierarchical HotNet.

	MultiFDRnet	FDRnet	HHotNet	HotNet2	Domino	Netmix2	BioNet
Simulation	3557(465)	2588(672)	3624(468)	3,740,063(18,900)	150(2)	44,493(122)	7382(1881)
Bladder cancer	3840	2079	3372	3,725,313	139	44,017	/
Head and neck cancer	2546	1852	3581	3,726,267	521	44,013	/

**Table 2 cancers-15-04090-t002:** Subnetworks identified by seven methods applied to bladder cancer and head and neck cancer data. hHotNet: Hierarchical HotNet. #subnetworks: the number of identified subnetworks; #genes: the total number of genes in identified subnetworks; # COSMIC genes: the number of genes that are included in the COSMIC cancer gene database.

	Bladder Cancer				Head and Neck Cancer		
Method	#Genes	#Subnetworks	FDR	#COSMIC Genes	#Genes	#Subnetworks	FDR
MultiFDRnet	77	24	0.084(0.01)	29	61	16	0.083(0.01)
FDRnet	95	28	0.086(0.01)	30	77	15	0.093(0.006)
hHotNet	22	1	0.028	17	33	2	0.06(0.05)
HotNet2	52	1	0.17	28	56	1	0.05
Domino	27	3	0.54(0.16)	10	110	15	0.50(0.18)
NetMix2	21	1	0.18	17	20	1	0.07
BioNet	/	/	/	/	/	/	/

## Data Availability

The bladder cancer and the head and neck cancer somatic mutation and copy number data (dbGaP study accession no. phs000178) were downloaded from the TCGA Firehose website (https://gdac.broadinstitute.org (accessed on 31 August 2022)). The iRefIndex18.0 PPI network, the BioGRID v4.4.212 PPI network, the ReactomeFI v2021 PPI network, the STRING 11.5 PPI network and the HNSC PPI network were downloaded from https://irefindex.vib.be/ (accessed on 25 August 2022), https://thebiogrid.org (accessed on 25 August 2022), https://reactome.org (accessed on 25 August 2022), https://string-db.org/ (accessed on 25 August 2022) and https://www.ndexbio.org/ (accessed on 31 August 2022), respectively, without any restriction.

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
