# Peer review of "Identifying Significantly Perturbed Subnetworks in Cancer Using Multiple Protein–Protein Interaction Networks"

_cancers, 2023, doi:10.3390/cancers15164090_

Round 1
Reviewer 1 Report
The manuscript introduced MultiFDRnet to detect significantly perturbed subnetworks utilizing multiple PPI networks. And the performance was compared with other methods by doing simulations and case studies. In general, this study is well-designed and well-written.
1. It is mentioned in Section 3.1 line 240-241 that the p-values for the target were randomly selected from a beta-distribution Beta(a, 1), and the p-values for non-target genes were sampled from uniform distribution. Can you explain why Beta and uniform distribution and why they are better than other probability distributions?
2. A range of beta distribution parameter a was used in the simulation but the author didn’t discuss how the performance/comparability change cross difference a in section 3.1.2.
3. Please supply the description relevant to how the Mixed-Integer Linear Programming was employed in both simulation and cancer data analysis.
Author Response
The manuscript introduced MultiFDRnet to detect significantly perturbed subnetworks utilizing multiple PPI networks. And the performance was compared with other methods by doing simulations and case studies. In general, this study is well-designed and well-written.
>>> We thank the reviewer for the encouraging comments. We have carefully revised the manuscript and addressed the reviewers’ comments point-by-point below. Changes in the resubmitted article are marked as blue text.
1. It is mentioned in Section 3.1 line 240-241 that the p-values for the target were randomly selected from a beta-distribution Beta(a, 1), and the p-values for non-target genes were sampled from uniform distribution. Can you explain why Beta and uniform distribution and why they are better than other probability distributions?
>>>We adopted the signal-to-noise decomposition model used in (Pounds and Morris, 2003, Dittrich et al., 2008) for the simulation study. It posits that the distribution of p-values consists of two parts: one stemming from the null hypothesis and the other from the alternative hypothesis. It is conceivable that p-values derived from the null model uniformly span the range (0,1). Under the alternative hypothesis, the distribution of p-values is characterized by a high density at lower values, which progressively diminishes as the p-values increase. This pattern aligns well with a specific form of the beta distribution: Beta(a,1). Therefore, we used Beta(a,1) for target genes and Uniform(0,1) for non-target genes in our study. In the revised paper, we added a description about the rationale behind the selection of these distributions. See Section 3.1, pp. 6, line 243-252.
Dittrich, et al. (2008) Identifying functional modules in protein–protein interaction networks: an integrated exact approach. Bioinformatics, 24: i223–i231.
Pounds, S. Morris (2003) Estimating the occurrence of false positives and false negatives in microarray studies by approximating and partitioning the empirical distribution of p-values. Bioinformatics, 19(10):1236-42.
2. A range of beta distribution parameter a was used in the simulation but the author did not discuss how the performance/comparability change cross difference a in Section 3.1.2.
>>> A smaller value of parameter a represents a stronger signal strength. With the decrease of signal strengths, we observed that (1) the performance of all the methods declined, and (2) in nearly all instances, the relative performance rankings of the different methods remained the same. This demonstrates the superiority of the proposed method over a wide range of signal strengths. We presented a brief discussion in the revised paper. See Section 3.1.2, pp. 7, line 296-299.
3. Please supply the description relevant to how the mixed-integer linear programming was employed in both simulation and cancer data analysis.
>>> Briefly, for both simulation and cancer data analyses, we started by selecting a set of seed genes. For each seed, a local graph was extracted by using the PageRank algorithm. We then formulated the problem of detecting a subnetwork in a local graph as a mixed-integer linear programming problem. We added a detailed discussion about how the mixed-integer linear programming was employed in Section 3.1, pp. 7, line 263-265, Section 3.2.1, pp. 10, line 389-390 and Section 3.3.2, pp. 12, line 448-452.
Reviewer 2 Report
In this paper, authors introduced MultiFDRnet, a novel methodology devised for the detection 472 of significantly perturbed subnetworks utilizing multiple PPI networks. MultiFDRnet addressed several limitations of existing network-based methods. The paper has technical depth and novel contribution. Written very well and structured with proper explanations. Only few theoretical explanation is missing. Other the paper good to be accepted in its current form.
NA
Author Response
The authors introduced MultiFDRnet, a novel methodology devised for the detection of significantly perturbed subnetworks utilizing multiple PPI networks. MultiFDRnet addressed several limitations of existing network-based methods. The paper has technical depth and novel contribution. Written very well and structured with proper explanations. Only few theoretical explanations are missing. The paper is good to be accepted in its current form.
>>> We thank the reviewer for the encouraging comments. We have carefully revised the manuscript and addressed the reviewers’ comments point-by-point below. Changes in the resubmitted article are marked as blue text. Following the suggestion, we added some theoretical explanations in the revised paper. See Section 2.3, pp. 4, line 139-144, line 147-150 and pp.5, line 188-190.
Reviewer 3 Report
The authors presented a computational approach named MultiFDRnet to identify significantly perturbed subnetworks by taking multiple PPI networks as input. The work is an extended work from their previously developed tool, FDRnet. MultiFDR’s improvement upon FDRnet is that MultiFDRnet was designed to address the limits of using one PPI network. The authors claimed that using one PPI network would suffer from (1) variations in the topological structures of individual networks and (2) incomplete structure of context-specific networks. The authors used simulation and experimental data to demonstrate MultiFDRnet’s efficacy. However, it has some significant issues listed below:
Major:
1. Domain-specific or context-specific networks seem more promising in specific research topics. The author didn’t show how their approach addressed the issue of the incomplete structure of context-specific networks.
2. The improvement of MultiFDRnet upon FDRnet is subtle (Fig 2c-d, Table 2). From a performance perspective, it’s concerning whether MultiFDRnet is not. The novelty of the approach is questionable as well.
3. The Methods section does not clearly define what “p-values from gene-level analysis” is.
4. How was the transition matrix constructed in the PPR random walk algorithm, given that PPI networks do not have weights? Specifically, how were the transition probabilities calculated for intra-layer nodes (genes in one layer) and inter-layer nodes (genes crossing layers)?
5. Only F-score measures were used to evaluate the performance in the simulation study, which do not provide a comprehensive comparison across different methods.
6. STRING has a combined network and individual networks, which could serve as an excellent dataset to test MultiFDRnet’s abilities in addressing the two gaps mentioned by the authors. Therefore, it is highly recommended to perform the following benchmarking experiments:
a. FDRnet/Other single PPI method + STRING combined network (as provided by STRING)
b. MultiFDRnet + all networks from STRING except the combined network.
It will also be exciting to see if excluding one network from MultiFDRnet increases the performance. This could support the authors’ point that individual networks might introduce noise in the results.
7. Robustness to randomness
It is not mentioned how much multiFDRnet relies on the reliability of individual networks. To prove this point, the authors could include one or more random or inaccurate networks as inputs and see how much it affects the performance.
Minor:
8. What does “control the FDRs of identified subnetwork” mean in line 301? And why is it important?
9. It’s unclear how AggrePPI was constructed. Please explain in detail.
The quality of the writing is good. Additional proofreading is needed.
Author Response
The authors presented a computational approach named MultiFDRnet to identify significantly perturbed subnetworks by taking multiple PPI networks as input. The work is an extended work from their previously developed tool, FDRnet. MultiFDR’s improvement upon FDRnet is that MultiFDRnet was designed to address the limits of using one PPI network. The authors claimed that using one PPI network would suffer from (1) variations in the topological structures of individual networks and (2) incomplete structure of context-specific networks. The authors used simulation and experimental data to demonstrate MultiFDRnet’s efficacy. However, it has some significant issues listed below:
>>>We have carefully revised the manuscript and addressed the reviewers’ comments point-by-point below. Changes in the resubmitted article are marked as blue text.
1. Domain-specific or context-specific networks seem more promising in specific research topics. The author did not show how their approach addressed the issue of the incomplete structure of context-specific networks.
>>> We agree with the reviewer that context-specific networks are more promising in specific research topics. Our method addresses the issue of the incomplete structure of context-specific networks by combining context-specific networks with general PPI networks to form a multiplex network, and this is a major novelty of the proposed method. The head and neck cancer study demonstrated that our method performed much better than existing methods. In the revised paper, we have further clarified this important point. See Section 1, pp. 2, line 77-79 and Section 3.3.2, pp. 12, line 438-443 and 448-452, pp. 13, line 469-482.
2. The improvement of MultiFDRnet upon FDRnet is subtle (Fig 2c-d, Table 2). From a performance perspective, it’s concerning whether MultiFDRnet is not. The novelty of the approach is questionable as well.
>>> We would respectfully disagree with the reviewer on this point. In Fig. 2c, we can see that, in terms of the ability to detect target genes as measured by F-scores, MultiFDRnet performed at least as well as FDRnet applied to individual PPI networks. In terms of the ability to detect target subnetworks as measured by Fsub scores (Fig. 2d), MultiFDRnet performed significantly better than FDRnet (p-value < 10^-5). We should emphasize that currently there is no consensus on which PPI networks should be used in real applications. Thus, it is significant to develop a method that can combine gene interaction information from multiple PPI networks. Please see Section 3.1.2, pp. 7, line 300-309, for details.
We observed similar results in the cancer studies. Briefly, for the bladder cancer study, FDRnet missed some well-known cancer genes. For the head and neck cancer study, our approach was the only method that successfully identified perturbed modular structures in a cancer-specific PPI network. Please see Sections 3.2.2. and 3.3.2 for a detailed discussion.
3. The Methods section does not clearly define what “p-values from gene-level analysis” is.
>>> We have now provided the definition. See Section 2, pp. 3, line 87-90.
4. How was the transition matrix constructed in the PPR random walk algorithm, given that PPI networks do not have weights? Specifically, how were the transition probabilities calculated for intra-layer nodes (genes in one layer) and inter-layer nodes (genes crossing layers)?
>>> We used the transition matrix of classic random walk in the PPR random walk algorithm in the study. Specifically, inter-layer transitions occur only between node pairs representing the same gene. Transition probabilities for both intra- and inter-layer node pairs are equal. We have provided a detailed description in the revised manuscript. See Section 2.3, pp. 5, line 193-201.
5. Only F-score measures were used to evaluate the performance in the simulation study. Why do not provide a comprehensive comparison across different methods?
>>> There may be some misunderstanding on this point. We did perform a comprehensive comparison. Specifically, we used three distinct metrics, namely, F-score, Fsub score, and FDR, to assess the abilities of an algorithm to detect target genes, to detect target subnetworks and to control FDRs, respectively. See Section 3.1.2, which we have now expanded, for a detailed description.
6. STRING has a combined network and individual networks, which could serve as an excellent dataset to test MultiFDRnet’s abilities in addressing the two gaps mentioned by the authors. Therefore, it is highly recommended to perform the following benchmarking experiments: (a) FDRnet/Other single PPI method + STRING combined network (as provided by STRING), (b) MultiFDRnet + all networks from STRING except the combined network.
>>> To clarify, the main goal of the study is as follows: there are four commonly used PPI networks, namely BioGRID, iRefIndex, ReactomeFI and STRING, each with a distinct topological structure. However, there is no consensus on which PPI networks should be used in real applications, and when a subnetwork detection method is applied to different PPI networks, detected cancer driver genes and subnetworks can be different. Therefore, it is significant to develop a new method that is capable of detecting significantly perturbed subnetworks by leveraging multiple PPI networks. The experiment presented in the paper has already sufficiently demonstrated that we have achieved this goal. The experiment suggested by the reviewer is interesting, but we feel that it is not appropriate within this study for the following reasons: (1) individual networks used in STRING (also called channels) represent different information resources. Since individual channels are rarely used, the experiment is not relevant to real applications. (2) The information from individual channels has different confidence levels. STRING combines channels in a probabilistic manner. For example, if 95% of interactions from a channel are false discoveries, the score assigned to the channel is very small. However, in our case, as we work on four commonly used PPI networks, we assume that they are equally good. (3) Some channels are databases and are not PPI networks. It is hard for us to combine them with other networks within our framework. We hope this clarifies the described strategy.
It will also be exciting to see if excluding one network from MultiFDRnet increases the performance. This could support the authors’ point that individual networks might introduce noise in the results.
>>> We did not claim that individual networks might introduce noise in the results. Please refer to the discussion presented above regarding the main purpose of the study.
7. It is not mentioned how much multiFDRnet relies on the reliability of individual networks. To prove this point, the authors could include one or more random or inaccurate networks as inputs and see how much it affects the performance.
>>> We thank the reviewer for this suggestion. We conducted an experiment to investigate the issue. Briefly, we first generated a random network for each of the four PPI networks, by randomly swapping the edges of each node while maintaining the original degree distributions. Then, we repeated the simulation study by replacing one of the four PPI networks by its random counterpart. We reported the experiment results in Section 3.1.3 and Appendix B. As we might expect, the performance of our method declined, however, it was very robust against variations from individual networks and still performed significantly better than all other approaches.
8. What does “control the FDRs of identified subnetwork” mean in line 301? And why is it important?
>>> False discovery rate (FDR) refers to the proportion of non-target genes within identified subnetworks. Essentially, a high FDR suggests that a large number of genes in a detected subnetwork actually are false alarms. Therefore, it is crucial to control the FDR to guarantee the accuracy and reliability of findings.
9. It is unclear how AggrePPI was constructed. Please explain in detail.
>>> Briefly, AggrePPI was constructed by aggregating all interactions presented in at least one of the original PPI networks. We provided a detailed description in the revised manuscript. See Section 3.1, pp. 6, line 234-235.
Reviewer 4 Report
Comment 1
Line 7 in simple summary, what FDR stands for in multiFDRnet in stead of appear in line 117
Line 9 what stands for PPI first appearance in stead of appear in line 41
line 55, 56 correct grammer as a full sentence
Author Response
We have carefully revised the manuscript and addressed the reviewers’ comments point-by-point below. Changes in the resubmitted article are marked as blue text.
1. Line 7: in simple summary, what FDR stands for in multiFDRnet instead of appear in line 117.
>>> Done. See Section Simple Summary, pp. 1, line 8.
2. Line 9: what stands for PPI first appearance instead of appear in line 41.
>>> Done. See Section Simple Summary, pp. 1, line 8-9.
3. Line 55, 56: correct grammar as a full sentence
>>>Done. See Section 1, pp. 2, line 56-58.